# Smoke-free prisons in England: indoor air quality before and after implementation of a comprehensive smoke-free policy

Leah R Jayes,[ORCID][1] Rachael L Murray,[1] Magdalena Opazo Breton,[1] Christopher Hill,[1] Elena Ratschen,[2] John Britton[1]

¹Division of Epidemiology and Public Health, University of Nottingham, Nottingham, UK
²Department of Health Sciences, University of York, York, UK

**Correspondence to**
Dr Leah R Jayes;
leah.jayes@nottingham.ac.uk

## ABSTRACT

**Objectives** High levels of particulate pollution due to secondhand smoke (SHS) have previously been recorded in English prisons. As part of an evaluation to ascertain whether a new comprehensive smoke-free policy introduced in the first four prisons in England was successfully implemented, this study compares indoor air quality on prison wing landing locations three months before and three months after going smoke-free.

**Design** An indoor air quality monitoring study, comparing SHS levels before and after a comprehensive smoke-free prison policy.

**Setting** The first four prisons in England to implement a comprehensive smoke-free policy.

**Primary and secondary measures** We compared concentrations of airborne particulate matter <2.5 microns in diameter ($PM_{2.5}$), as a marker for SHS, on wing landing locations three months before and three months after the smoke-free policy was implemented. Static battery operated aerosol monitors were used to sample concentrations of $PM_{2.5}$ on wing landings.

**Results** After discarding data from monitors that had been tampered with we were able to analyse paired data across four prisons from 74 locations, across 29 wing landing locations, for an average sampling time of five hours and eight minutes. When comparing samples taken three months before with the paired samples taken three months after policy implementation (paired for prison, day of the week, time of day, wing location and position of monitor), there was a 66% reduction in mean $PM_{2.5}$ concentrations across the four prisons sampled, from 39 to 13 µg/m³ (difference 26 µg/m³, 95% CI 25 to 26 µg/m³).

**Conclusion** Prison smoke-free policies achieve significant improvements in indoor air quality. A national smoke-free policy would therefore be an effective means of protecting prisoners and staff from harm due to SHS exposure in the prison environment.

## INTRODUCTION

Since it was introduced a decade ago, smoke-free legislation in the UK has been successful in protecting the general public and workforce from harm arising from exposure to second-hand smoke (SHS).[1–3] However,

### Strengths and limitations of this study

► This is the first study to compare particulate pollution before and after the implementation of a smoke-free policy in English prisons.
► Air quality monitoring was not carried out blind, it is possible that prisoners and staff may have changed their behaviour during data collection.
► Pre-policy samples were taken during the winter months and post-policy samples were taken during the summer months, greater ventilation post-policy may have contributed to the reduction in particulate matter.

the legislation included an exemption for Her Majesty's Prison and Probation Service (HMPPS, formally The National Offender Management Service (NOMS)) in England and Wales.[4] The exemption allowed prisoners aged over 18 years to smoke in a single cell or in a cell shared with other smokers,[5] staff smoking was prohibited within prison perimeter walls. Since around 80% of UK prisoners are smokers,[6–9] and many of these are highly tobacco dependent,[9 10] prisoners and prison staff remain at risk of high levels of SHS exposure.

In recent years, HMPPS has come under mounting pressure, from both the Prison Officers' Association (the trade union representing prison officers throughout the UK)[11] and from legal challenges by non-smoking prisoners citing poor health due to personal frequent exposure to SHS,[12] to implement a smoke-free policy throughout the prison estate in England and Wales. In September 2015, in response to empirical research demonstrating high levels of SHS in English prisons,[13 14] HMPPS announced the pilot implementation of a comprehensive smoke-free policy in four prisons in the South-West of England.[15] This policy prohibited all staff

members and prisoners from smoking tobacco and possessing tobacco or smoking paraphernalia (such as lighters and cigarette rolling paper) within the perimeter walls of the four prisons.

Prior to implementation, smoking cessation services were offered to prisoners free of charge (this included behavioural support and pharmacotherapy), and disposable electronic cigarettes were made available to purchase through the prison canteen. Prisoners were only permitted to use an electronic cigarette whilst in their cell. Tobacco and smoking paraphernalia were removed from the canteen list two weeks before the smoke-free date at each establishment, to give prisoners the opportunity to smoke but not replace any remaining tobacco before the implementation date.

As part of an evaluation of the smoke-free policies introduced in four prisons in the South-West of England in 2016, we have compared indoor air quality, measured as concentrations of airborne particulate matter <2.5 microns in diameter ($PM_{2.5}$), on wing landing locations three months before and three months after going smoke-free. By measuring concentrations of $PM_{2.5}$, this study intends to determine to what extent the new policy reduces concentrations of SHS.

## METHODS
### Study prisons
Data were collected from the first four English Her Majesty's Prison (HMP) Service establishments selected to go smoke-free, with one prison going smoke-free every two weeks between 11 April and 23 May 2016. The prisons were all in the South-West of England, and were selected for reasons including their low transfer rate to other regional areas, being all-male establishments, and having a relatively stable population. According to HMPPS annual performance ratings, all four prisons were performing well at the time of data collection and had reported no recent incidents.[16] One was a local prison (HMP 1) which served the courts and held both remand and convicted prisoners, while the other three were training prisons (HMPs 2, 3, and 4) which only held sentenced prisoners who are likely to be employed in day time activities (eg, workshops or education). All four prisons had a Care and Separation Unit.

Before the smoke-free policy was implemented, all four prisons had a non-smoking policy for staff members within the perimeter wall, while prisoners were allowed to smoke only in their cells. However, although not permitted, prisoner smoking still occurred regularly on the exercise yards. The only exception prior to the smoke-free policy was the residential healthcare unit at HMP 1, which was designated 'smoke-free' and in which all indoor smoking was prohibited, to include cells. This unit was therefore excluded from our study.

### Particulate pollution
The concentration of airborne particulate matter <2.5 microns in diameter ($PM_{2.5}$) is a well-established marker of

indoor SHS concentrations,[17 18] and previous studies have shown high $PM_{2.5}$ concentrations in environments where smoking has taken place.[17 19] Battery operated SidePak Personal Aerosol Monitors AM510 (TSI, Minnesota, USA) have been successfully used to measure $PM_{2.5}$ in prison environments previously,[14 20–22] as they are small, portable and do not require mains electricity (giving researchers the freedom over static placement on the wing landings). The SidePak uses a built-in sampling pump to draw air through the device, which then measures the concentration in milligrams per cubic metre of $PM_{2.5}$. The monitor logs $PM_{2.5}$ measurements at one minute intervals, with each one minute data point being an average of 60 one second sample measurements. Eleven SidePak monitors fitted with impactor heads in order to measure $PM_{2.5}$ and set to a calibration factor of 0.30, as appropriate for tobacco smoke,[23 24] were used to measure $PM_{2.5}$ concentrations at each prison visit for this study. In accordance with manufacturer's instructions, SidePak devices were cleaned, the impactor re-greased, zero-calibrated and the flow rate set at 1.7 L/min before each use. Data were collected over three or four consecutive weekdays before the smoke-free policy was introduced. Where possible, data collection was then repeated after the policy was introduced (repeating data collection at the same prison, day of the week, time of day, wing location and position of monitor). See table 1 for study prison characteristics and data collection dates.

Two researchers trained in the use of air quality monitors placed the SidePak monitors in static locations on wing landings. Samples collected were compared with current World Health Organisation (WHO) indoor air quality standards, which recommend that $PM_{2.5}$ concentrations alone should not exceed $25\,\mu g/m^3$ as a 24-hour mean.[25]

### Data collection
The four prisons were visited by two researchers three months before and after each prison's smoke-free implementation date (see table 1). The two researchers were assigned a prison officer during their data collection to gain access to all the wing landings to place the SidePak air monitors in static locations, and to advise on areas of the prison that were not currently accessible for the researchers to visit (typically due to prisoner incidents). A wing landing is the communal shared area that all cell doors on a wing open onto, often housing showers, telephones and is typically a place where prisoners can spend time out of their cell during designated periods of the day. Pre-policy, researchers aimed to gain access to every prison landing at all four prisons at least once to sample $PM_{2.5}$ concentrations. Each air quality sample was identified with a unique code and data were recorded by a researcher on a sampling log sheet, to include; the prison; date and day of data collection; wing location and position of monitor; time the monitor was switched on and off; whether there was evidence that the monitor had been moved or tampered with; monitor serial number,

**Table 1** Study prison characteristics and data collection dates

| Prison | HMP 1 | HMP 2 | HMP 3 | HMP 4 |
|---|---|---|---|---|
| Category and function* | Male Category B Local | Male Category C Training | Male Category C Training | Male Category C Training |
| Structural design | Built 1850s Victorian radial design | Built early 1800s Singular wings | Built 1974 Five two story living blocks and quick build wings | Built 1960s Mix of triangular, T-shaped and quick build wings |
| No of wings | 7 | 7 | 9 | 9† |
| Smoke-free implementation date | 11/04/16 | 25/04/16 | 9/05/16 | 23/05/16 |
| Prisoner roll count pre-policy | 505 | 634 | 706 | 518 |
| Prisoner roll count post-policy | 477 | 628 | 691 | 378† |
| Sampling dates pre-policy | 19/01/16 – 23/01/16 | 08/02/16 – 11/02/16 | 15/02/16 – 18/02/16 | 29/02/16 – 02/03/16 |
| Sampling dates post-policy | 05/07/16 – 08/07/16 | 18/07/16 – 21/07/16 | 22/08/16 – 25/08/16 | 15/08/16 – 17/08/16 |

*Category B prisons hold prisoners for whom the very highest conditions of security are not necessary but for whom escape must be made very difficult. Category C prisons hold prisoners who cannot be trusted in open conditions but who do not have the resources and will to make a determined escape attempt. Local prisons serve the courts and receive remand and post-conviction prisoners prior to their allocation to other establishments. Training prisons hold sentenced prisoners who tend to be employed in a variety of activities such as prison workshops, gardens and education and in offending behaviour programmes.
†HMP 4 closed two wings (and transferred all prisoners located on these wings) between pre- and post-smoke-free data sampling dates.
HMP, Her Majesty's Prison.

and visit number (visit 1=pre-policy implementation, visit 2=post-policy implementation). Typically, the two researchers were escorted around each prison twice a day, (morning and afternoon) in order to retrieve and place monitors in static locations. Researchers worked as a pair, with one completing the sampling log sheet while the other positioned or retrieved the monitors and checked if they had been tampered with or moved. Pre-implementation sampling logs and unique codes were used post-implementation to guide repeat data collection; where feasible placing SidePak monitors on the same day of the week, wing location, monitor position, start time and duration of sample. The sampling duration of each dataset was determined by access to wings locations via the prison escort and the machine's battery life (around 11 hours). Monitors were programmed to turn off before the end of their battery life. The monitors were usually placed halfway down the wing, above head height and away from open outside doors, windows or cooking equipment. Where possible, monitors were placed in discreet static locations to avoid disrupting prisoners' normal behaviour. For security reasons, researchers advised the officers on each wing how long they should expect the monitor to stay on the landing for and where each monitor had been placed.

As air quality monitors had been removed by prisoners during earlier sampling at HMP 2, all monitors in this prison were placed at one end of the unit next to or inside the wing office. Therefore, samples were not directly taken from the wing landings. Due to the landing design of several wings at HMP 3, air quality monitors had to be placed in a cupboard which inhibited air flow.

### Patient and public involvement
There was no patient or public involvement in this study.

### Data analysis
Each dataset was downloaded from the SidePak device using the monitor's recommended software (Trakpro V.4.6.1) and imported into STATA V.13, alongside its unique code. Datasets were then paired using their unique code (paired for prison, day of the week, wing and monitor position) and corresponding sample times paired (to the minute) to compare $PM_{2.5}$ concentrations pre-implementation and post-implementation. Data from monitors that appeared to have been moved or tampered with, and those with no paired sample, were discarded. Descriptive statistics for all paired data and paired data by prison ID were generated; including mean, range, median, IQR and the proportion of time the $PM_{2.5}$ concentration exceeded WHO 24-hour mean $PM_{2.5}$ upper limit of $25\,\mu g/m^3$.[25] The percentage change of $PM_{2.5}$ concentrations was determined by comparing the mean and median $PM_{2.5}$ levels overall and in each prison before and after smoke-free. The Wilcoxon signed-rank test was used to assess statistical significance between pre-implementation and

post-implementation PM₂.₅ concentrations in each establishment. To illustrate the sampled PM₂.₅ distribution from each prison before and after implementation of smoke-free, box plots were constructed. Although PM₂.₅ data distributions were skewed, we present arithmetic mean figures throughout since these are used by WHO to define their upper guidance limits.[25]

## RESULTS

A total of 200 datasets were collected from 32 wing landing locations throughout the four prisons. One SidePak monitor was destroyed during pre-implementation data collection, and on 12 occasions monitors were tampered with by prisoners (eg, by blocking the air inlet or turning off the monitor). The remaining 187 datasets included 113 collected before and 74 collected after policy implementation; the lower number after implementation arose primarily from restrictions on access to some prison wings. We therefore generated 74 paired sets of data for analysis (paired by prison, day of the week, time of the day, wing and monitor placement) which are presented in this paper. The 74 paired data sets were taken from across 29 wing landings (post-policy two wings at HMP 4 had been closed and one wing at HMP 3 could not be accessed by researchers due to security concerns), sampling particulate matter for an average of 5 hours and 8 minutes. Across all four prisons, monitors were placed on wing landings in the morning between 8:16 and 10:22 and in the afternoon between 14:38 and 18:00. (See table 2 for individual prison break down of mean sampling times and monitor placements times). On sampling days both pre-policy and post-policy implementation, all wings (apart from the Care and Separation Units) were at or near full capacity, with prisoner occupancy per wing ranging from 19 to 180.

### Combined data from all four prisons, comparing PM₂.₅ concentrations collected pre-implementation and post-implementation

Mean PM₂.₅ concentrations on wing landing locations before the introduction of smoke-free policy were 39 μg/m³, and 13 μg/m³ after introduction, representing a 66% reduction in mean PM₂.₅ concentrations (mean difference 26 μg/m³, 95% CIs 25 to 26 μg/m³); and a 69% reduction in median PM₂.₅ concentrations (from 26 to 8 μg/m³). The mean PM₂.₅ concentration pre-implementation exceeded WHO 24-hour mean PM₂.₅ upper limit of 25 μg/m,[25] and continuously monitored levels were above this limit for half of all sampling time (see table 2).

### Individual data from all four prisons, comparing PM2.5 concentrations collected pre-implementation and post-implementation

Data for the four prisons sampled (table 2) demonstrate that all but HMP 2 had mean PM₂.₅ concentrations above WHO 24-hour mean upper limit pre-policy implementation, and all had mean post-policy concentrations below this limit.[25] All four prisons saw a statistically significant reduction in the PM₂.₅ concentration pre- to post- smoke-free policy (median percentage reductions, HMP 1=81%, HMP 2=45%, HMP 3=67%, HMP 4=72%, all four prisons, p<0.001). HMP 1, the local prison, had the highest mean and median PM₂.₅ concentrations pre-policy, and the largest percentage reduction post-policy for these samples. In HMP 2 (where monitors were not placed directly on the wing landings), the time spent over WHO 24-hour mean PM₂.₅ upper limit reduced from 7% to 0%; in the other three prisons, the reduction was from 53%–77% pre-policy to 13%–15% post-policy implementation. Figure 1 shows box plots of the distribution of PM₂.₅ concentrations measured in each prison before and after the smoke-free policy. An example of the difference in PM₂.₅ concentration profiles on a main residential wing at HMP 3, pre-implementation and post-implementation is presented in figure 2.

## DISCUSSION

This is the first study to compare particulate pollution before and after the implementation of smoke-free policy in English prisons. The air quality measures, which used concentrations of PM₂.₅ as a proxy for SHS, demonstrate that before the smoke-free policy was introduced PM₂.₅ levels were well in excess of WHO 24-hour mean PM₂.₅ upper limit,[25] with half of all sampling time over this recommended guidance level. After introduction of the smoke-free policy, there was a substantial and statistically significant reduction in PM₂.₅ concentrations, to below WHO upper guidance limit of 25 μg/m³ per 24 hours. However, the range of concentrations sampled suggests that prisoners were still smoking on occasions under the smoke-free policy.

Our air quality measurements were not carried out in blind fashion, because researchers were obliged to answer questions from staff members and prisoners who enquired about the monitoring. However, while it is possible that prisoners or staff changed their behaviour in response to being monitored, we think that is unlikely to have occurred to any appreciable degree over the course of our measurements. SHS is not the only source of indoor PM₂.₅, which includes particulate matter released from sources such as open fires, toasters and microwaves. However, where toasters and microwaves were present on the wings, every effort was made to place the SidePak monitors as far away from these as possible. Safe locations for the SidePak monitors were limited, but researchers tried to collect data from all wings at each prison. Since security concerns and the design of the wings at HMP 2 and HMP 3 required us to place the SidePak monitors in wing offices (not directly on the wing landing) and in cupboards on several of the landings (which inhibited air flow), these measures are likely to have underestimated the true PM₂.₅ concentrations on these wing locations pre- and post-smoke-free policy. Nevertheless, reductions in PM₂.₅ concentrations were still observed after policy implementation in the majority of these samples. Similar issues with placement of SidePak monitors on wing locations were described in work carried out in a New Zealand prison, but

**Table 2** Summary of sampled PM$_{2.5}$ concentrations combined and individually for four prisons pre- and post- smoke-free implementation

| Visit* | Combined data | | HMP 1 | | HMP 2 | | HMP 3 | | HMP 4 | |
|---|---|---|---|---|---|---|---|---|---|---|
| | 1 | 2 | 1 | 2 | 1 | 2 | 1 | 2 | 1 | 2 |
| No of paired datasets (total paired sample time hour:min) | 74 380:20 | | 20 97:57 | | 14 70:57 | | 22 125:50 | | 18 85:36 | |
| Mean sample time (hour:min) | 5:08 | | 4:54 | | 5:04 | | 5:43 | | 4:45 | |
| Range of sampling start times (hour:min) | 8:16–10:22 14:38–18:00 | | 8:16– 9:02 14:57– 18:00 | | 8:52– 9:22 15:39– 15:54 | | 8:42–9:57 14:38–16:14 | | 9:08–10:20 15:22–16:29 | |
| Arithmetic mean (and 1min range) of PM$_{2.5}$ concentrations (µg/m³) | 39 (0–1359) | 13 (0–3073) | 66 (2–678) | 14 (0–635) | 13 (0–121) | 6 (0–30) | 35 (0–1359) | 15 (2–227) | 36 (1–1058) | 17 (0–3073) |
| Arithmetic mean percentage reduction from pre-implementation to post-implementation | 66% | | 79% | | 50% | | 58% | | 54% | |
| Median (and IQR) of PM$_{2.5}$ concentrations (µg/m³) | 26 (15–46) | 8 (4–15) | 42 (27–76) | 8 (4–16) | 11 (6–17) | 6 (2–9) | 27 (17–44) | 9 (5–17) | 29 (18–44.5) | 8 (4–18) |
| Median percentage reduction from pre-implementation to post-implementation | 69% | | 81% | | 45% | | 67% | | 72% | |
| Percentage of time above 25 µg/m³† | 51% | 11% | 77% | 13% | 7% | 0% | 53% | 14% | 56% | 15% |

*Visit number, 1=pre-smoke-free policy implementation, 2=post-smoke-free policy implementation.
†WHO 24 hour mean PM$_{2.5}$ upper limit of 25 µg/m³.
HMP, Her Majesty's Prison.

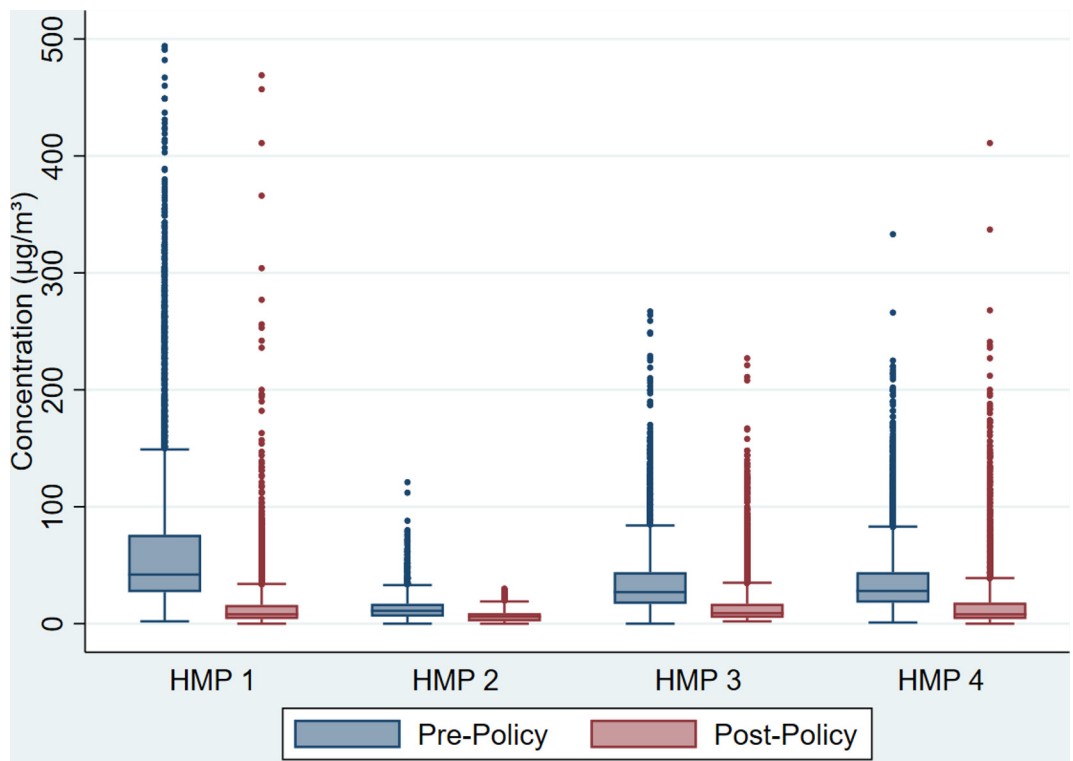

**Figure 1** Box plots of PM₂.₅ distributions in each of the four prisons pre- and post-smoke-free implementation. The horizontal line in each box represents the median value and the top and bottom of the box represent the 25th and 75th percentile, with the lines extending from the top and bottom of the boxes widening to the 5th and 95th percentile of the distribution. For ease of representation, figure 1 does not show $PM_{2.5}$ concentrations over $500\,\mu g/m^3$ (this only applies to samples taken from HMPs 1 and 4). HMP, Her Majesty's Prison.

that study also reported a significant reduction in $PM_{2.5}$ concentration after going smoke-free.[22]

As an inevitable consequence of the smoke-free implementation dates in the four prisons, pre-policy air quality samples were taken during the winter months and post-policy during the summer months. It is possible that greater ventilation through open windows in the summer months may have contributed to the reduction in particulate levels between these two time points. However, to minimise this

bias, SidePak monitors were placed towards the centre of the wings and away from any open windows during sampling. To examine whether outdoor air pollution (not only derived from SHS) could have contributed to indoor $PM_{2.5}$ concentrations, a study which measured concentrations of particulate matter in 15 Scottish prisons, compared its indoor $PM_{2.5}$ concentrations to outdoor measurements taken via the nearest static government monitoring station.[21] Unfortunately, for this study, the nearest static government monitors

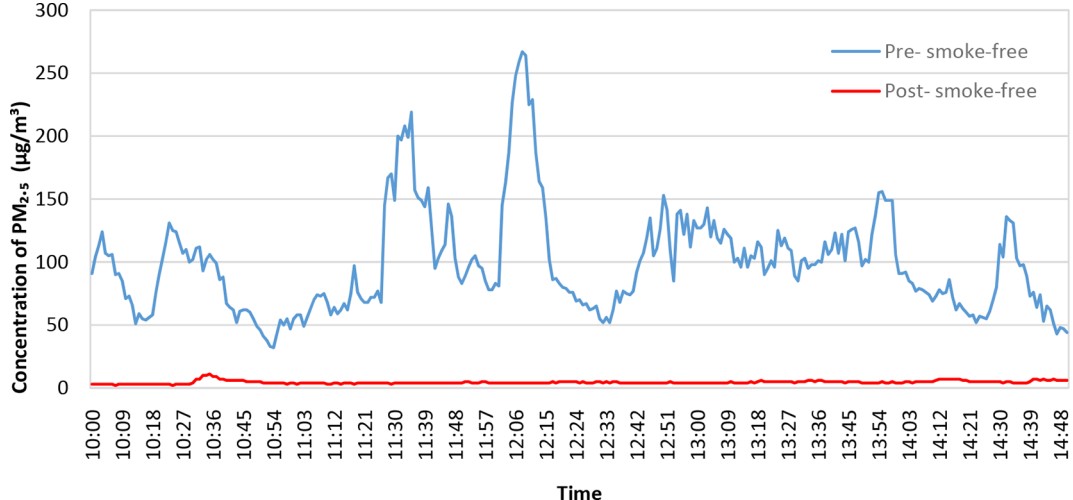

**Figure 2** Concentrations of $PM_{2.5}$ sampled on a main residential wing at HMP 3 pre- and post- smoke-free implementation. HMP, Her Majesty's Prison.

were a considerable distance away (mean, 47 km) from the four prison sites sampled and were all placed in urban city centre locations (three of the four prisons sampled in this study were in remote rural locations). As PM2.5 is not specific to SHS and can also arise from traffic and industrial air pollution, researchers felt the comparison for this study was not suitable.

Pre-policy, researchers were able to work around any prison incidents (eg, regime changes, prisoner disturbances) in an attempt to sample all wing locations throughout the four prisons. Post-policy, researchers did not have the same flexibility as the sampling schedule was predetermined (in order to pair the samples for prison, day of the week, time, wing and monitor location), therefore fewer datasets were collected. We are unable to say whether this has significantly biased our findings. We recognise that our estimates of the proportion of time spent above WHO PM2.5 upper guidance limit of 25 µg/m³ as 24-hour mean are not truly representative because the maximum sampling time was determined by access to the wings, the battery life of the SidePak monitors used (around 11 hours), and only being able to place the monitors onto the wings during daytime hours. Since smoking does not occur during sleep, particulate levels are likely to have been considerably lower during the night. However, our data give a very good estimation, in view of the large amount of paired data collected pre-policy and post-policy (over 15 days pre-policy and post-policy), of SHS pollution during times when non-smokers would be exposed during waking hours.

In an earlier air quality monitoring study (which included two of the pilot smoke-free sites sampled here), we measured PM2.5 concentrations on wing landings where prisoners were permitted to smoke in their cells that were slightly higher than those three months prior to the smoke-free implementation in the present study (mean values 44 µg/m³ and 39 µg/m³, respectively).[14] A possible explanation for this is that the majority of samples taken in the current study were carried out on days leading up to the weekly delivery of tobacco to prisoners from the prison shop (data in this study were collected Monday to Friday, with canteen delivery typically occurring on Fridays) when many prisoners are running out of tobacco, whereas the earlier study included samples taken at the weekend (after tobacco delivery). This earlier study reported that PM2.5 concentrations were higher immediately after canteen delivery days.[14] It is also possible however that three months before going smoke-free, prisoners were already starting to reduce their tobacco consumption or had been on a smoking cessation course at the prison in light of the impending policy. Further validation of SHS levels recorded in this study pre-policy comes from two further air quality monitoring studies carried out in Scottish prisons prior to their smoke-free policy[13 21] which produced similar pre-policy PM2.5 concentrations.

Since 2005, the USA, New Zealand, Canada and Australia have all implemented smoke-free policies in their correctional facilities. International air quality studies from New Zealand and the USA have shown that comprehensive prison smoke-free policies are effective in substantially reducing SHS concentrations.[20 22 26] All of these studies used markers of SHS, respirable particulate matter (eg, particulate matter less than 2.5 µg/m³ (PM2.5))[20 22] and airborne nicotine[26] to sample prison locations pre-policy and post-policy. The percentage reductions in PM2.5 concentrations in our study were very similar to those recorded in these other countries.[20 22 26]

Alongside reduced SHS concentrations, the potential health benefits of introducing a comprehensive smoke-free policy have been outlined in a study which examines the 10 years since the USA implemented its smoking ban in prisons. This study found that prisons which implemented a smoke-free policy had a 9% reduction in smoking-related deaths (particularly cardiovascular and pulmonary deaths), and that bans in place for longer than 9 years were associated with a reduction in cancer deaths.[27] A study exploring natural deaths in male prisoners over 60 years of age in England and Wales reported diseases of the circulatory system (such as, coronary heart disease and cerebrovascular disease) and respiratory illnesses, all of which are substantially more common among smokers, as the most common cause of death.[28] With these findings in mind, the roll-out of a comprehensive smoke-free policy across all 121 prisons in England and Wales set out by HMPPS[15] has the potential to have the same positive health impact on the nearly 83 000 prisoners[29] currently held, and the 32 000 staff members employed,[30] in the prison estate.

Findings from this study suggest that prisoners were still smoking after the introduction of smoke-free, since PM2.5 concentrations post-policy ranged from 0 to 3073 µg/m³, consistent with continued smoking in some areas. Hammond and Emmons measured nicotine concentrations before and after prisons in California, USA went smoke-free, and concluded that a smoking ban was effective in reducing SHS exposure but did not eliminate it.[26] An ethnographic case study conducted in 10 prisons in the USA after implementing a complete smoking ban described the lengths prisoners would go to in order to acquire, exchange and smoke tobacco, and how tobacco had now become a more lucrative commodity to sell due to big demand and higher profit margin than illicit drugs.[31] The study concluded that although prisoners smoked less post-policy, the emergent black market created by banning tobacco had a negative impact on prisoners. The emergence of a tobacco black market was also observed in New Zealand and the Northern Territories of Australia after their implementation of smoke-free.[31–33]

## CONCLUSION

Smoke-free policies in these prisons have successfully reduced prisoner smoking, and both prisoner and staff exposure to SHS. Further work to reduce still further the occurrence of prisoner smoking is clearly required, and to assess the impact of the smoke-free policy on prisoner health. However, our data provide strong evidence in support of the continued implementation of the smoke-free policy throughout the

English and Welsh prison estate and in other penal systems internationally.

**Acknowledgements** The lead author would like to thank all four prisons for agreeing to participate in this research study, alongside the staff members who escorted the researchers during data collection.

**Contributors** LJ is the guarantor and takes responsibility for the integrity of the work as a whole, from inception to publication; study conception and design, data collection, analysis of data, interpretation of data and drafting the research manuscript. RM, ER and JB contributed to the study conception and design. CH assisted with data collection and MOB supported data preparation in STATA and data analysis. RM and JB aided drafting the manuscript. All authors read and approved the final manuscript.

**Funding** This work was supported by Medical Research Council (grant number MR/K023195/1) for the UK Centre for Tobacco and Alcohol Studies, which includes funding from the British Heart Foundation, Cancer Research UK, Economic and Social Research Council and the Department of Health under the auspices of the UK Clinical Research Collaboration.

**Disclaimer** The funders had no role in the study design, data collection and analysis, decision to publish or preparation of the manuscript.

**Competing interests** None declared.

**Patient consent for publication** Not required.

**Ethics approval** University of Nottingham, Medical School Ethics Committee (G06062013 CHS EPH) approved this study, it was then subsequently approved by HMPPS, National Research Committee (NRC) (Ref: 2013–202) in July 2014. Permission to enter all four prisons for data collection was sought from the Deputy Director of Public Sector Prisons and the Deputy Director of Custody for the South-West area. The Governors at each prison also agreed to the research being undertaken at their establishments.

**Provenance and peer review** Not commissioned; externally peer reviewed.

**Data sharing statement** All available data can be obtained by contacting the corresponding author.

**Author note** Researchers (LJ and CH) were security cleared to Enhanced Level 1, enabling them to visit establishments and work within HMPPS. Researchers were given security talks from each establishments prior to data collection.

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
