## [Reviewer comments · BMJ Open]

ARTICLE DETAILS

TITLE (PROVISIONAL)	Smoke-free prisons in England: Indoor air quality before and after implementation of a comprehensive smoke-free policy
AUTHORS	Jayes, Leah; Murray, Rachael; Opazo Breton, Magdalena; Hill, Christopher; Ratschen, Elena; Britton, John

VERSION 1 - REVIEW

REVIEWER	Sean Semple University of Stirling, UK
REVIEW RETURNED	16-Aug-2018

GENERAL COMMENTS	This is an excellent paper that describes a series of comprehensive measurements taken before and after implementation of smoke-free restrictions in the first of four prisons in England as part of a wider roll out. It provides important findings showing the substantive improvements in air quality that were experienced in these establishments as a result of this policy change, and these results are important both at national level to inform wider roll-out and internationally as more countries consider similar restrictions to smoking in prisons. I commend the authors for gathering such a large amount of data in clearly difficult settings. I have some comments I would ask the authors to consider and have divided these in to major and minor. Major 1. The manuscript needs to provide detail about the duration of measurements. Currently this is only presented in a limited and oblique fashion. The devices used had a battery life of 11h and that devices were only placed in a position during 'daylight hours'. From table 2 it is possible to make some calculation of average duration of sampling (though this is confused further by the listing of 148 pairs – should this be 74 pairs of measurements? – and so a calculation of average duration of sampling being either ~ 5h or 10h. I would like to see a much more explicit description of sampling durations at each prison (mean, range etc) for both pre and post (and paired). 2. Linked to this comment the discussion mentions as a weakness that measurements were not carried out for a full 24h period and states 'as smoking doesn't take place during sleep PM levels are likely to be lower during the night'. While this is, to an extent, true an 11h period (say from 10am to 9pm) may miss out on the two peaks of prisoner smoking (just as they go to sleep and as soon as they wake up in the morning). In this sense the timing of the sampling is vitally important to the results generated. It would be important for the revised paper to present the mean duration and
---

the mean start time of sampling for pre and post measurements. Only by doing this can we be confident that the pre and post data are directly comparable.

3. Another point on this topic. Were the measurements made until the battery on the Sidepak failed? It is not considered good practice to use data from a device where the battery is failing/running low. Sometimes this can lead to the pump producing low flow for the final minutes. If this is the case it may be prudent to exclude the final 15-30 minutes of the data collected.

4. The discussion rightly talks about a possible seasonal effect in terms of ventilation and also recognises that PM2.5 is not specific to SHS. Given these two points is it not also the case that changes in outdoor ambient air concentration of PM2.5 could account for some (or even all) of the changes in measured PM2.5 concentrations. It should be possible to check for changes in outdoor PM2.5 concentrations quite easily by looking at the nearest government outdoor PM2.5 monitor to each of the four prisons and to see what the average concentrations were on the days of the measurements performed. Similar analysis was carried out for a study of 15 prisons in Scotland and is available as a report via a link at the bottom of this page
http://www.sps.gov.uk/Corporate/News/Creating_a_Smoke_Free_Prison_Environment.aspx
 It is worth noting that outdoor ambient PM2.5 concentrations around these 15 Scottish prisons had a median of 9 ug/m3. A similar approach to account for the contribution of ambient air pollution may produce much greater SHS-derived PM2.5 reductions (for example if we look at HMP1 as an example and assume 9ug/m3 of PM2.5 was from non-SHS sources at both time points then we would see a reduction from 30ug/m3 (pre) to 4 ug/m3 (post). This equates to an 87% reduction rather than a 66% reduction.

5. Lastly, I would like to have seen some comparison to other UK data measuring PM2.5 in prisons. As mentioned above there is a study and paper by Semple et al., 2017 Ann Work Expo Health. 2017 Aug 1;61(7):809-821. Doi: 10.1093/annweh/wxx058 that presented PM2.5 data collected for approximately 6 days continuously in all 15 Scottish prisons. The median value of that data (32 ug/m3) is very similar to these results and thus supports the validity of these findings. Similarly there is a report "Semple S, Galea K, Walsh P et al. (2015a) Report on Second-Hand Smoke in Prisons: Final Report. 3513889A. Available at <https://www.gov.uk/government/publications/air-qualityreports>. " that provides useful comparative data from six English prisons that should be discussed given that those data were gathered prior to smoke-free policies being widely discussed.

Minor

1. Abstract. Consider providing some indication of measurement periods in the results. Also suggest presenting the concentration results to two significant figures (so a reduction from 39 to 13 ug/m3). The figures to 0.01 ug/m3 are pretty meaningless and are beyond the unit of quantification of the device used. This also applies to the presentation of the results in the main body of the paper.

2. Page 4. Final sentence states that the study 'intends to determine whether the policy is sufficient in reducing concentrations of SHS'. Suggest thinking if 'sufficient' is the best term here? What would be sufficient – a 20% reduction, a 50% reduction, a 99% reduction?

	3. Table 2. Row labelled 'Number of paired datasets'. I find this confusing – the total adds to 148 but I believe this should be 74 in total (i.e. half). Currently if we assume the total matched time is spread over the number of paired datasets then we get an average of approximately 5h of sampling on each pre and post occasion. I think this should be nearer 10h. 4. Table 2. Row labelled 'Arithmetic mean (and range of PM2.5 concentrations. Is this the 1-min range or the sampling duration average range for each measurement location? It looks like it must be the 1-min average but this should be made explicit. 5. Table 2. HMP2 % of time >25ug/m3. This is 1% for the pre-policy measurements suggesting this prison was already essentially smoke-free before the policy was implemented. Worth discussing (and excluding data from this prison in the analysis?) 6. Conclusions. I would like to see a separate section of conclusions within the discussion. The final paragraph of page 16 states that prisoners are still smoking. I think it is worth pointing out that the frequency has considerably reduced as evidenced by the major reductions in the % of sampling minutes when readings were >25 ug/m3. Sean Semple 16th August 2018
--	---

REVIEWER	Sam McCrabb University of Newcastle, Australia
REVIEW RETURNED	01-Nov-2018

GENERAL COMMENTS	This is a well written paper which assess air quality in UK prisons following smoke-free policy implementation. While there is only one aim and result from this study, it is novel and important to this field of research. 1. Page 6, line 30. Said prison were selected as no prisoner incidents? 2. Until discussion, it is unclear how long monitors were left, if they were collected etc. I believe it is worth making this clearer in the data collection section, especially given the WHO 24-hour mean upper limit is used as cut off. 3. HMP 1 is referred to as "the local prison" what does this mean? I understand the other prisons are training prisons however I find this term confusing. 4. Figure 2 is small and unclear, suggest updating 5. Could list avenues for other potential work at end of discussion but not essential, e.g. staff perception of policy implementation on incarcerated persons. 6. Was there any smoking cessation support provided prior to policy implementation to incarcerated persons? Policy may have impacted cessation rates. Were they incarcerated persons provided with any other forms of nicotine delivery devices to use (such as electronic cigarettes and are these covered under the ban)?
--

REVIEWER	Helen Sweeting University of Glasgow, UK
REVIEW RETURNED	05-Nov-2018

GENERAL COMMENTS

Comments for the authors

Thank you for asking me to review this paper which reports on PM2.5 measurements via SidePak monitors in the wing landings of four prisons in South-West England approx three months before and then three months after implementation of a complete smoking ban. The paper adds to the currently very small evidence base on the impact of prison smoking restrictions internationally, and although my comments look extensive, they are mainly fairly small and should be easily dealt with – most are around clarification of methods.

ABSTRACT

Obj – suggest ‘... to ascertain whether A [replacing THE] new comprehensive ...’ – otherwise it sounds a bit as if all readers should know/understand about the E & W policy.

Measures – this uses words to repeat info on three months before/after which is in Objectives, but doesn’t say how measurements were made (SidePak monitors) which means the ‘After discarding data from MONITORS ...’ in the Results para comes as a bit of a surprise.

STRENGTHS / LIMITATIONS

First bullet –typo - cut ‘THE’ before ‘English’.

Second bullet – should HMPPS acronym be spelled out? Typo – ‘for its success ..’. And this suggests that measurmeents in these four prisons provided evidence for continued roll-out of the smoke-free policy – but would the roll-out actually have been stopped on the basis of these measurements? My understanding was that the Sept 2015 Andrew Selous letter described a comitment to roll-out throughout E & W (“Two recent academic studies commissioned by NOMS have identified that high levels of second hand smoke in some communal areas are still prevalent in some prisons. ... The findings of these studies have reinforced our commitment to move towards smoke free prisons as soon as possible in a safe and controlled way.”)

INTRODUCTION

P4 (of 24) 1st para – probably worth clarifying here that HMPPS is E & W – so readers don’t assume it’s UK and the reference to ‘the prison estate in E & W’ in 2nd para makes sense.

P4, 2nd para – not completely clear what the policy was at the time (ie who could smoke and exactly where), since this was the context of the ‘before’ measures.

METHODS

P5, 1st para – suggest cutting ‘The’ and amending to ‘ONE WAS A local prison HMP1), HOLDING ...’

P5, 1st para – ‘traiNing establishments’ sounds like something specialist – in fact it’s not (I found the explanation in Table 1) – it would be good to have this in the text.

P5, 2nd para – think first sentence belongs earlier (relates to comment re Intro) and should clarify that prisoners were allowed to smoke outdoors.

P5, 3rd para – I'd welcome more on the SidePak monitors – what are these? Size/ portability? How do they work? What's a PM2.5 impactor? Are there validity/reliability data? What was the reason for choosing these over other measurement methods?

P6, 1st para – please clarify '... each one minute data point being an average of 60 secs of sample measurements' – maybe I'm being a bit dim, but I don't understand how it could really be anything other than that.

P6, 1st para – Needs some clarification - 'staff escort availability' initially suggested to me that the monitors were carried around (ie mobile measurements) – but actually I think it was just staff to escort researchers to the landings. Were they exactly the same places before and after? And how long in each location – the manuscript notes an 11 hour battery life, but I didn't know whether that meant the monitors were placed for 11 hours in each location (and if not, why not – and who decided)?

P6, 2nd para – again needs some clarification. Did the two researchers go in together? Did they aim to measure once on every landing in each prison? Were the monitors hung up (some seem to have been placed quite high) or on shelves/tables? 'Officers on each wing were advised where each monitor had placed and for how long' – who advised them? Why did the officers need to know how long the monitor had been there? At some point it's probably also important to explain what a 'wing landing' is, for readers who don't know.

P6, 2nd para – typo – 'The sampling duration of each DATASET WAS determined ...'.

P6, last line – suggest '... had been REMOVED by prisoners ...'.

P7 – this info re repeat data collection probably belongs earlier.

P9, 1st para – is Trackpro software something specific to SidePak? And re '... monitors that appeared to have been moved or tampered with ...' – I'm wondering if this needs a sentence in data collection – who checked this – was it the researchers returning and finding the monitor not quite how they'd left it?

P9, 1st para – were Wilcoxon's tests (ie non-parametric) used because the data were skewed?

P9, 1st para – typo – '... alongside ITS unique code'.

RESULTS

P10, 1st para – '... collected from 29 wing landings ...' - Table 2 shows total wings as 32 (with two closed on one prison, a bit unclear whether these are included on the table), so does this mean that data couldn't be collected from three wings? And it's only at this stage, when I read about 113 datasets before and 74 after that I realise that these represent several days in all/most wings.

	P10, 1st para – which prison had the CSU? P11 – Table 2 – I don't think the 51%/11% in the final row can be correct – this is adding the other percentages, but shouldn't it be their average? Otherwise, if the others were all – say – 30%, by this logic, this row would read 120%, which doesn't make sense. P12 – 'Excluding HMP 2 ..' – but should it be excluded? Isn't this a bit like saying 'excluding the one which already happened to be low, the others reduced', but HMP 2 is as much part of the sample as the others. P12 ref to Fig 2 – unfortunately the reproduction in the paper is tiny, with poor resolution, so I couldn't check this. DISCUSSION P14, 1st para – '... half of all sampling time ...' – think this relates to the 51% as per my comment above, so amend if this is a mistake. P15, 1st para – '... limited by regime changes or episodes of lock-down due to prisoner incidents, which resulted in fewer samples after the policy ...' – this seems to be implying that there were more incidents post implementation – is this correct? Is it an issue/worth commenting on? Re 11 hour sampling limitation – still unclear – did this mean that researchers put it somewhere then returned the next day (by which time battery flat) to replace battery and move it? And given the various limitations listed, I'm wondering whether saying the data gives a 'VERY good estimation' might be a slight over-statement? P16, 2nd para – typo – 'found that prisons WHICH implemented..'. ' P16, 2nd para, final sentence – suggest '... roll-out of A COMPREHENSIVE smoke-free POLICY across and the 32,000 staff ...' P16, final para – 0-3073 – is this high figure correct? The box plots suggest this would be a massive outlier. P17, final sentence – 'evidence in support of extending smoke-free policy throughout the E & W ...' – same comment as on this point in the strengths/limitations bullets.
--	--

VERSION 1 – AUTHOR RESPONSE

Reviewer: 1

Reviewer Name: Sean Semple

Institution and Country: University of Stirling, UK. Please state any competing interests or state 'None declared': None declared

This is an excellent paper that describes a series of comprehensive measurements taken before and after implementation of smoke-free restrictions in the first of four prisons in England as part of a wider roll out. It provides important findings showing the substantive improvements in air quality that were

experienced in these establishments as a result of this policy change, and these results are important both at national level to inform wider roll-out and internationally as more countries consider similar restrictions to smoking in prisons.

I commend the authors for gathering such a large amount of data in clearly difficult settings.

I have some comments I would ask the authors to consider and have divided these in to major and minor.

Major

1. The manuscript needs to provide detail about the duration of measurements. Currently this is only presented in a limited and oblique fashion. The devices used had a battery life of 11h and that devices were only placed in a position during 'daylight hours'. From table 2 it is possible to make some calculation of average duration of sampling (though this is confused further by the listing of 148 pairs – should this be 74 pairs of measurements? – and so a calculation of average duration of sampling being either ~ 5h or 10h. I would like to see a much more explicit description of sampling durations at each prison (mean, range etc) for both pre and post (and paired).

2. Linked to this comment the discussion mentions as a weakness that measurements were not carried out for a full 24h period and states 'as smoking doesn't take place during sleep PM levels are likely to be lower during the night'. While this is, to an extent, true an 11h period (say from 10am to 9pm) may miss out on the two peaks of prisoner smoking (just as they go to sleep and as soon as they wake up in the morning). In this sense the timing of the sampling is vitally important to the results generated. It would be important for the revised paper to present the mean duration and the mean start time of sampling for pre and post measurements. Only by doing this can we be confident that the pre and post data are directly comparable.

Please see below the response to both of these comments.

Apologies there seems confusion over the data being comparable and number of paired datasets.

This paper only reports results from paired data (pre to post policy) using data being paired for prison, day of the week, location (including wing and static monitor placement on the wing) and time of day (akin to paired air quality data in Wilson et al, 2012, REFRESH study). Making it directly comparable. There were 74 paired datasets (148 individual data sets) used for the analysis and reported on in this paper. Data with no corresponding paired sample (pre to post) was discarded and not reported in this paper. The following additions/amendments have been made to make this clearer under the following headings.

Abstract

Results

'...When comparing samples taken three months before with the paired samples taken three months after policy implementation (paired for prison, day of the week, time of day, wing location and position of monitor),'

Methods

Particulate pollution

'Data were collected over three or four consecutive weekdays before the smoke-free policy was introduced, where possible, data collection was then repeated after the policy was introduced (repeating data collection at the same prison, day of the week, time of day, wing location and position of monitor).'

Data analysis

'Datasets were then paired using its unique code (pairing prison, day of the week, wing and monitor position) and corresponding sample times paired (to the minute) to compare PM_{2.5} concentrations pre- and post-implementation.'

Results

'We therefore generated 74 paired sets of data for analysis (paired by prison, day of the week, time of day, wing and monitor placement) which are presented in this paper.'

Average sampling duration was indeed around 5 hours. Overall, the average sample time for all 74 paired datasets was 5 hours and 8 minutes, we have also calculated these for each individual prison (all data added into Table 2).

Mean sample time (hr:min)	5:08	4:54	5:04	5:43	4:45
------	------	------	------	------

Also included the overall average sampling time to the body of the text under 'Results'. As follows:

Results

'The 74 paired data sets were taken from across 29 wing landings (post-policy two wings at HMP 4 had been closed and one wing at HMP 3 could not be accessed by researchers due to security concerns), sampling particulate matter for an average of 5 hours and 8 minutes'

In terms of adding in the mean start time of sampling, this may not be so appropriate for the whole dataset given that there were two sampling periods during each day, with monitors being placed on wing locations in the morning (time range 08:16 – 10:22) and late afternoon (time range 14:38 – 18:00). Therefore, we feel that giving one mean start time would not truly reflect the data sampled. Instead, we have provided more detail on the two sampling periods and have provided ranges of the start times for the morning and afternoon sampling start time for each prison, provided in Table 2 (alongside the mean sampling times).

Mean sample time (hr:min)	5:08	4:54	5:04	5:43	4:45
Range of sampling start times (hr:min)	AM: 08:16 – 10:22 PM: 14:38 – 18:00	AM: 08:16 - 09:02 PM: 14:57 - 18:00	AM: 08:52 - 09:22 PM: 15:39 – 15:54	AM: 08:42 – 09:57 PM: 14:38 - 16:14	AM: 09:08- 10:20 PM: 15:22- 16:29

The following text has been amended to reflect this.

Methods

Data collection

'Typically, the two researchers were escorted around each prison twice a day, (morning and afternoon) in order to retrieve and place monitors in static locations.'

Results

'Across all four prisons, monitors were placed on wing landings in the morning between 08:16 – 10:22 and in the afternoon between 14:38-18:00. (See Table 2 for individual prison break down of mean sampling times and monitor placements times).'

3. Another point on this topic. Were the measurements made until the battery on the Sidepak failed? It is not considered good practice to use data from a device where the battery is failing/running low. Sometimes this can lead to the pump producing low flow for the final minutes. If this is the case it may be prudent to exclude the final 15-30 minutes of the data collected.

No, measurements were not made until the battery on the SidePak failed. For those monitors which were left in the prison to run into the evening, the researchers programmed them to turn off before the end of their battery life. Added sentence to clarify.

Data collection

Monitors were programmed to turn off before the end of their battery life.

4. The discussion rightly talks about a possible seasonal effect in terms of ventilation and also recognises that PM2.5 is not specific to SHS. Given these two points is it not also the case that changes in outdoor ambient air concentration of PM2.5 could account for some (or even all) of the changes in measured PM2.5 concentrations. It should be possible to check for changes in outdoor PM2.5 concentrations quite easily by looking at the nearest government outdoor PM2.5 monitor to each of the four prisons and to see what the average concentrations were on the days of the measurements performed. Similar analysis was carried out for a study of 15 prisons in Scotland and is available as a report via a link at the bottom of this page

http://www.sps.gov.uk/Corporate/News/Creating_a_Smoke_Free_Prison_Environment.aspx

It is worth noting that outdoor ambient PM2.5 concentrations around these 15 Scottish prisons had a median of 9 ug/m3. A similar approach to account for the contribution of ambient air pollution may produce much greater SHS-derived PM2.5 reductions {for example if we look at HMP1 as an example and assume 9ug/m3 of PM2.5 was from non-SHS sources at both time points then we would see a reduction from 30ug/m3 (pre) to 4 ug/m3 (post). This equates to an 87% reduction rather than a 66% reduction.

This is a really interesting comment. Having looked into this recommendation further (via www.airqualityengland.co.uk / <https://uk-air.defra.gov.uk/latest/currentlevels>), please see the table below regarding nearest outdoor government monitors and their readings during the same sampling times pre and post smoke-free per prison.

	Nearest government outdoor monitor measuring PM2.5	Distance from prison	Type of location	Mean PM2.5 on pre-smoke-free sampling dates (ug/m3)	Mean PM2.5 on post-smoke-free sampling dates (ug/m3)
HMP 1 City centre location	Saltash Callington Road	37 miles	Urban traffic	16.1	4 (data missing for full duration)
HMP 2 Rural location	Saltash Callington Road	14 miles	Urban traffic	10	6.2
HMP 3 Rural location	Plymouth centre	38 miles	Urban traffic	11.2	7.2
HMP 4 Rural location	Bristol St Pauls	26 miles	Urban traffic	5.8	12.4

The table demonstrates that most prisons were a considerable distance from the nearest outdoor PM_{2.5} monitoring station (mean, 29 miles) with all sampling city centre locations. Three out of the four prisons sampled in this study were remote rural locations. We therefore feel these data are unlikely to be representative of the outdoor air quality of the majority of the sampled prison sites. A section has been added to the discussion to reflect this, as follows:

'To examine whether outdoor air pollution (not only derived from SHS) could have contributed to indoor PM_{2.5} concentrations, a study which measured concentrations of particulate matter in 15 Scottish prisons, compared its indoor PM_{2.5} concentrations to outdoor measurements taken via the nearest static government monitoring station [21]. Unfortunately, for this study, the nearest static government monitors were a considerable distance away (mean, 47 kilometres) from the four prison sites sampled and were all placed in urban city centre locations (three of the four prisons sampled in this study were in remote rural locations). As PM_{2.5} is not specific to SHS and can also arise from traffic and industrial air pollution, researchers felt the comparison for this study was not suitable.'

5. Lastly, I would like to have seen some comparison to other UK data measuring PM_{2.5} in prisons. As mentioned above there is a study and paper by Semple et al., 2017 Ann Work

Expo Health. 2017 Aug 1;61(7):809-821. Doi: 10.1093/annweh/wxx058 that presented

PM_{2.5} data collected for approximately 6 days continuously in all 15 Scottish prisons. The median value of that data (32 ug/m³) is very similar to these results and thus supports the validity of these findings. Similarly there is a report "Semple S, Galea K, Walsh P et al.

(2015a) Report on Second-Hand Smoke in Prisons: Final Report. 3513889A. Available at

<https://www.gov.uk/government/publications/air-qualityreports>. "that provides useful comparative data from six English prisons that should be discussed given that those data were gathered prior to smoke-free policies being widely discussed.

Apologies, reference to both of these studies have now been included in the discussion section. As follows:

'Further validation of SHS levels recorded in this study pre-policy comes from two air quality monitoring studies carried out in Scottish prisons prior to their smoke-free policy [13, 21] which produced similar pre-policy PM_{2.5} concentrations.'

Minor

1. Abstract. Consider providing some indication of measurement periods in the results. Also suggest presenting the concentration results to two significant figures (so a reduction from

39 to 13 ug/m³). The figures to 0.01 ug/m³ are pretty meaningless and are beyond the unit of quantification of the device used. This also applies to the presentation of the results in the main body of the paper.

Agree, now added in the mean measurement period in the Abstract, under Results. As follows:

'After discarding data from monitors that had been tampered with we were able to analyse paired data across four prisons from 74 locations, across 29 wing landing locations, for an average sampling time of five hours and eight minutes.'

Also, we have amended all concentrations of PM_{2.5} to two significant figures for all data presented throughout the paper.

2. Page 4. Final sentence states that the study 'intends to determine whether the policy is sufficient in reducing concentrations of SHS'. Suggest thinking if 'sufficient' is the best term here? What would be sufficient – a 20% reduction, a 50% reduction, a 99% reduction?

Now removed 'sufficient' to read as follows:

'By measuring concentrations of PM_{2.5}, this study intends to determine to what extent the new policy reduces concentrations of SHS.'

3. Table 2. Row labelled 'Number of paired datasets'. I find this confusing – the total adds to

148 but I believe this should be 74 in total (i.e. half). Currently if we assume the total matched time is spread over the number of paired datasets then we get an average of approximately 5h of sampling on each pre and post occasion. I think this should be nearer 10h.

There were 74 paired data sets in total, with an average paired sampling time of around 5 hours. Hopefully by addressing the comments above this is now clearer.

4. Table 2. Row labelled 'Arithmetic mean (and range of PM2.5 concentrations. Is this the 1- min range or the sampling duration average range for each measurement location? It looks like it must be the 1-min average but this should be made explicit.

This is the 1 minute range. Have now added in '1 minute range' to make this clearer.

5. Table 2. HMP2 % of time >25ug/m3. This is 1% for the pre-policy measurements suggesting this prison was already essentially smoke-free before the policy was implemented. Worth discussing (and excluding data from this prison in the analysis?)

As outlined in 'Data collection', unfortunately due to devices previously being removed by prisoners during the first day of sampling, we had to place the monitors in/next to the wing office so staff could ensure monitors were not tampered with. This section has been amended to make this clearer, now as follows:

'As air quality monitors had been removed by prisoners during earlier sampling at HMP 2, all monitors were placed at one end of the unit next to or inside the wing office. Therefore, samples were not directly taken from the wing landings.'

Essentially, even though we were not able to sample directly off the wing landing for HMP 2, it still valuable, in that, there was still a significant reduction in PM2.5 levels pre- to post- smoke-free policy. Therefore we feel it is important not to exclude this data, simply due to samples not being directly taken from the wing landing. This is now discussed further in the discussion section.

'Since security concerns and the design of the wings at HMP 2 and HMP 3 required us to place the Sidepak monitors in wing offices (not directly on the wing landing) and in cupboards on several of the landings (which inhibited air flow), these measures are likely to have underestimated the true PM_{2.5} concentrations on these wing locations pre- and post- smoke-free policy. Nevertheless, reductions in PM_{2.5} concentrations were still observed after policy implementation in the majority of these samples.'

6. Conclusions. I would like to see a separate section of conclusions within the discussion. The final paragraph of page 16 states that prisoners are still smoking. I think it is worth pointing out that the frequency has considerably reduced as evidenced by the major reductions in the % of sampling minutes when readings were >25 ug/m3.

Now added in a conclusion at the end of the discussion, as follows:

'Conclusion

Smoke-free policies in these prisons has successfully reduced prisoner smoking, and both prisoner and staff exposure to SHS. Further work to reduce still further the occurrence of prisoner smoking is clearly required, and to assess the impact of the smoke-free policy on prisoner health. However, our data provide strong evidence in support of the continued implementation of the smoke-free policy throughout the English and Welsh prison estate and in other penal systems internationally. '

Sean Semple 16th August 2018

Reviewer: 2

Reviewer Name: Sam McCrabb

Institution and Country: University of Newcastle, Australia. Please state any competing interests or state 'None declared': None declared

This is a well written paper which assess air quality in UK prisons following smoke-free policy implementation. While there is only one aim and result from this study, it is novel and important to this field of research.

1. Page 6, line 30. Said prison were selected as no prisoner incidents?

This refers to the researchers gaining access to the prison wings for Sidepak placement, the assigned prison officer would advise on areas of the prison throughout the visit that were not accessible (or safe) for the researchers to visit which was often due to a prisoner incident. The wording of this section has been amended in an attempt to make this clearer.

'The two researchers were assigned a prison officer during their data collection to gain access to all the wing landings to place the SidePak air monitors, and to advise on areas of the prison that were not currently accessible for the researchers to visit (typically due to prisoner incidents).'

2. Until discussion, it is unclear how long monitors were left, if they were collected etc. I believe it is worth making this clearer in the data collection section, especially given the WHO 24-hour mean upper limit is used as cut off.

Overall, the average sample time for all 74 paired datasets was 5 hours and 8 minutes which is now outlined in Table 2 alongside the average sample time per prison (also in Table 2). The overall average sampling time has also been added to the body of the text under 'Results'. As follows:

Results

'The 74 paired data sets were taken from across 29 wing landings (post-policy two wings at HMP 4 had been closed and one wing at HMP 3 could not be accessed by researchers due to security concerns), sampling particulate matter for an average of 5 hours and 8 minutes'

Alongside this, information has been added on the two periods of sampling during each day, with monitors being placed on wing locations in the morning (time range 08:16 – 10:22) and late afternoon (time range 14:38 – 18:00). The range of sampling start times has been added in Table 2 (alongside the mean sampling time).

The following text has been amended to reflect this.

Methods

Data collection

'Typically, the two researchers were escorted around each prison twice a day, (morning and afternoon) in order to retrieve and place monitors in static locations.'

Results

'Across all four prisons, monitors were placed on wing landings in the morning between 08:16 – 10:22 and in the afternoon between 14:38-18:00. (See Table 2 for individual prison break down of mean sampling times and monitor placements times).'

3. HMP 1 is referred to as "the local prison" what does this mean? I understand the other prisons are training prisons however I find this term confusing.

Apologies, this may be a UK specific term. There is a key under Table 1 for 'Category and function*'.

*Local prisons serve the courts and receive remand and post-conviction prisoners prior to their allocation to other establishments.

This definition has also been added to the body of the text under 'study prisons'. Amended text as follows:

One was a local prison (HMP 1) which served the courts and held both remand and convicted prisoners, while the other three were training prisons (HMPS 2, 3, & 4) which only held sentenced prisoners who are likely to be employed in day time activities (e.g. workshops or education).

4. Figure 2 is small and unclear, suggest updating

Figures 1 and 2 are currently provided as per requested under BMJ formatting guidelines (<https://authors.bmj.com/writing-and-formatting/formatting-your-paper/>)

Currently in TIFF format with resolution of 600 dpi. Please could BMJ Open advise if this format needs amending.

5. Could list avenues for other potential work at end of discussion but not essential, e.g. staff perception of policy implementation on incarcerated persons.

As highlighted below, avenues for further work outlined in the discussion:

'Further work to reduce still further the occurrence of prisoner smoking is clearly required, and to assess the impact of smoke-free policy on prisoner health'

6. Was there any smoking cessation support provided prior to policy implementation to incarcerated persons? Policy may have impacted cessation rates. Were they incarcerated persons provided with any other forms of nicotine delivery devices to use (such as electronic cigarettes and are these covered under the ban)?

Yes. Please see amended section from the Introduction below providing more detail:

'Prior to implementation, smoking cessation services were offered to prisoners free of charge (this included behavioural support and pharmacotherapy), and disposable electronic cigarettes were made available to purchase through the prison canteen. Tobacco and smoking paraphernalia were removed from the canteen list two weeks before the smoke-free date at each establishment, to give prisoners the opportunity to smoke but not replace any remaining tobacco before the implementation date.'

Reviewer: 3

Reviewer Name: Helen Sweeting

Institution and Country: University of Glasgow, UK. Please state any competing interests or state 'None declared': None declared

Comments for the authors

Thank you for asking me to review this paper which reports on PM_{2.5} measurements via SidePak monitors in the wing landings of four prisons in South-West England approx three months before and then three months after implementation of a complete smoking ban. The paper adds to the currently very small evidence base on the impact of prison smoking restrictions internationally, and although my comments look extensive, they are mainly fairly small and should be easily dealt with – most are around clarification of methods.

ABSTRACT

1. Obj – suggest '... to ascertain whether A [replacing THE] new comprehensive ...' – otherwise it sounds a bit as if all readers should know/understand about the E & W policy.

Agreed, amended as suggested:

'evaluation to ascertain whether a new comprehensive smoke-free'

2. Measures – this uses words to repeat info on three months before/after which is in Objectives, but doesn't say how measurements were made (SidePak monitors) which means the 'After discarding data from MONITORS ...' in the Results para comes as a bit of a surprise.

Added a sentence to 'Primary and secondary measures' to make this clearer. Now reads as follows:

'Static battery operated aerosol monitors were used to sample concentrations of PM_{2.5} on wing landings.'

STRENGTHS / LIMITATIONS

3. First bullet –typo - cut 'THE' before 'English'.

Amended and removed the word 'the'. Now reads as follows:

'This is the first study to compare particulate pollution before and after the implementation of a smoke-free policy in English prisons.'

4. Second bullet – should HMPPS acronym be spelled out? Typo – 'for its success ...'. And this suggests that measurements in these four prisons provided evidence for continued roll-out of the smoke-free policy – but would the roll-out actually have been stopped on the basis of these measurements? My understanding was that the Sept 2015 Andrew Selous letter described a commitment to roll-out throughout E & W ("Two recent academic studies commissioned by NOMS have identified that high levels of second hand smoke in some communal areas are still prevalent in some prisons. ... The findings of these studies have reinforced our commitment to move towards smoke free prisons as soon as possible in a safe and controlled way.")

At the request of the editor, this bullet point has now been removed as it does not refer to a strength or limitation of the study.

INTRODUCTION

5. P4 (of 24) 1st para – probably worth clarifying here that HMPPS is E & W – so readers don't assume it's UK and the reference to 'the prison estate in E & W' in 2nd para makes sense.

Have added 'England and Wales' to clarify. Now reads as follows:

'However the legislation included an exemption for Her Majesty's Prison and Probation Service (HMPPS, formally The National Offender Management Service (NOMS)) in England and Wales (4).'

6. P4, 2nd para – not completely clear what the policy was at the time (ie who could smoke and exactly where), since this was the context of the 'before' measures.

Have added detail of the policy for clarification. Now reads:

'The exemption allowed prisoners aged over 18 years to smoke in a single cell or in a cell shared with other smokers [5], staff smoking was prohibited within prison perimeter walls.'

METHODS

7. P5, 1st para – suggest cutting 'The' and amending to 'ONE WAS A local prison HMP1), HOLDING ...'

Amended as suggested. Amended text reads as follows:

'One was a local prison (HMP 1) which served the courts and held both remand and convicted prisoners, while the other three were training prisons (HMPS 2, 3, & 4) which only held sentenced prisoners who are likely to be employed in day time activities (e.g. workshops or education).'

8. P5, 1st para – 'trainiNg establishments' sounds like something specialist – in fact it's not (I found the explanation in Table 1) – it would be good to have this in the text.

Amended text to outline what local and training prisons are. Amended text as follows:

'One was a local prison (HMP 1) which served the courts and held both remand and convicted prisoners, while the other three were training prisons (HMPS 2, 3, & 4) which only held sentenced prisoners who are likely to be employed in day time activities (e.g. workshops or education).'

9. P5, 2nd para – think first sentence belongs earlier (relates to comment re Intro) and should clarify that prisoners were allowed to smoke outdoors.

We hope the amended sentence (see above) in the introduction is now clearer about the legislation pre- smoke-free policy therefore have kept this sentence where it is.

Pre-smoke-free, prisoners were not allowed to smoke outdoors, however they did. Have amended this section in an attempt to make this clearer too. Amended sentence reads as follows:

'Before the smoke-free policy was implemented, all four prisons had a non-smoking policy for staff members within the perimeter wall, while prisoners were allowed to smoke only in their cells. However, although not permitted, smoking still occurred regularly on the exercise yards.'

10. P5, 3rd para – I'd welcome more on the SidePak monitors – what are these? Size/ portability? How do they work? What's a PM2.5 impactor? Are there validity/reliability data? What was the reason for choosing these over other measurement methods?

Added additional detail to this section on how and why the SidePak monitors were used.

Amended section reads as follows:

'Battery operated SidePak Personal Aerosol Monitors AM510 (TSI Inc, MN, USA) have been successfully used to measure PM_{2.5} in prison environments previously (14, 20-22), as they are small, portable and do not require mains electricity (giving researchers the freedom over static placement on the wing landings). The SidePak uses a built-in sampling pump to draw air through the device, which then measures the concentration in milligrams per cubic metre of PM_{2.5}. The monitor logs PM_{2.5} measurements at one minute intervals, with each one minute data point being an average of 60 one seconds sample measurements. Eleven SidePak Monitors AM510 fitted with impactor heads in order to measure PM_{2.5} and set to a calibration factor of 0.30, as appropriate for tobacco smoke (23, 24) were used to measure PM_{2.5} concentrations at each prison visit for this study.'

11. P6, 1st para – please clarify '... each one minute data point being an average of 60 secs of sample measurements' – maybe I'm being a bit dim, but I don't understand how it could really be anything other than that.

This sentence has been amended to make this clearer. Reads as follows:

'The monitor logs PM_{2.5} measurements at one minute intervals, with each one minute data point being an average of 60 one seconds sample measurements'.

12. P6, 1st para – Needs some clarification - 'staff escort availability' initially suggested to me that the monitors were carried around (ie mobile measurements) – but actually I think it was just staff to escort researchers to the landings. Were they exactly the same places before and after? And how long in each location – the manuscript notes an 11 hour battery life, but I didn't know whether that meant the monitors were placed for 11 hours in each location (and if not, why not – and who decided)?

The 'staff escort availability' has been removed and amended 'Abstract' and 'Methods' to make it clear that these were static measurements. Also added in the 'Methods' section how researchers moved around the prison to sample and how they tried to repeat data collected pre-policy, post-policy.

Abstract

Primary and secondary outcomes

'Static battery operated aerosol monitors were used to sample concentrations of PM_{2.5} on wing landings.'

Results

'...When comparing samples taken three months before with the paired samples (paired for prison, day of the week, time of day, wing location and position of monitor) taken three months after policy implementation,'

Methods

Particulate pollution

'Data were collected over three or four consecutive weekdays before the smoke-free policy was introduced, where possible, this data collection was then repeated after the smoke-free policy was introduced (repeating data collected at the same prison, wing and monitor location, weekday and time). Two researchers trained in the use of air quality monitors placed the SidePak monitors in static locations on wing landings.'

Particulate pollution

'Data were collected over three or four consecutive weekdays before the smoke-free policy was introduced, where possible, data collection was then repeated after the policy was introduced (repeating data collection at the same prison, day of the week, time of day, wing location and position of monitor). See Table 1 for study prison characteristics and data collection dates.

Two researchers trained in the use of air quality monitors placed the SidePak monitors in static locations on wing landings.'

Data collection

Typically, the two researchers were escorted around each prison twice a day, (morning and afternoon) in order to retrieve and place monitors in static locations. Researchers worked as a pair, with one completing the sampling log sheet whilst the other positioned or retrieved the monitors and checked if they had been tampered with or moved.

'Where possible, monitors were sat in discreet static locations to avoid disrupting prisoners' normal behaviour.'

Data analysis

Datasets were then paired using its unique code (paired for prison, day of the week, wing and monitor position) and corresponding sample times paired (to the minute) to compare PM_{2.5} concentrations pre- and post-implementation.'

Overall, the average sample time for all 74 paired datasets was 5 hours and 8 minutes which is now outlined in Table 2 alongside the average sample time per prison (also in Table 2).

Mean sample time (hr:min)	5:08	4:54	5:04	5:43	4:45
Range of sampling start times (hr:min)	AM: 08:16 – 10:22 PM: 14:38 – 18:00	AM: 08:16 - 09:02 PM: 14:57 - 18:00	AM: 08:52 - 09:22 PM: 15:39 – 15:54	AM: 08:42 – 09:57 PM: 14:38 - 16:14	AM: 09:08- 10:20 PM: 15:22- 16:29

The overall average sampling time has also been added to the body of the text under 'Results'. As follows:

'Results

The 74 paired data sets were taken from across 29 wing landings (post-policy two wings at HMP 4 had been closed and one wing at HMP 3 could not be accessed by researchers due to security concerns), sampling particulate matter for an average of 5 hours and 8 minutes.'

Alongside this, information has been added on the two periods of sampling during each day, with monitors being placed on wing locations in the morning (time range 08:16 – 10:22) and late afternoon (time range 14:38 – 18:00). The range of sampling start times has been added in Table 2 (alongside the mean sampling time (see above section from Table 2)).

The following text has been amended to reflect this.

Methods

Data collection

'Typically, the two researchers were escorted around each prison twice a day, (morning and afternoon) in order to retrieve and place monitors in static locations.'

Results

'Across all four prisons, monitors were placed on wing landings in the morning between 08:16 – 10:22 and in the afternoon between 14:38-18:00. (See Table 2 for individual prison break down of mean sampling times and monitor placements times).'

13. P6, 2nd para – again needs some clarification. Did the two researchers go in together? Did they aim to measure once on every landing in each prison? Were the monitors hung up (some seem to have been placed quite high) or on shelves/tables? 'Officers on each wing were advised where each monitor had placed and for how long' – who advised them? Why did the officers need to know how long the monitor had been there? At some point it's probably also important to explain what a 'wing landing' is, for readers who don't know.

Clarification on how researchers collected the data and worked in pairs added. Reads as follows under 'Data collection':

'The two researchers were assigned a prison officer during their data collection to gain access to all the wing landings to place the SidePak air monitors in static locations, and to advise on areas of the prison that were not currently accessible for the researchers to visit (typically due to prisoner incidents).'

'Typically, the two researchers were escorted around each prison twice a day, (morning and afternoon) in order to retrieve and place monitors in static locations. Researchers worked as a pair, with one completing the sampling log sheet whilst the other positioned or retrieved the monitors and checked if they had been tampered with or moved. Pre-implementation sampling logs and unique codes were used post-implementation to guide repeat data collection; where feasible placing SidePak monitors on the same day of the week, wing location, monitor position, start time and duration of sample.'

Have added a sentence about aiming to sample all prison wings pre-policy:

'Pre-policy, researchers aimed to gain access to every prison landing at all four prisons at least once to sample PM_{2.5} concentrations.'

Monitors were not hung up during sampling. Added that they sat in discreet locations, reads as follows:

'Where possible, monitors were sat in discreet static locations to avoid disrupting prisoners' normal behaviour.'

Sentence made clearer on who advised the officers of the monitor placement and why. Now reads:

'For security reasons, researchers advised the officers on each wing how long they should expect the monitor to stay on the landing for and where each monitor had been placed.'

Finally, added a sentence describing what a wing landing is, under 'Data collection' first paragraph.

'A wing landing is the communal shared area that all cell doors on a wing open onto, often housing showers, telephones and is typically a place where prisoners can spend time out of their cell during designated periods of the day.'

14. P6, 2nd para – typo – 'The sampling duration of each DATASET WAS determined ...'.

Amended typo to read as follows:

'The sampling duration of each datasets was determined by access to wings locations in order to collect monitors and the machines battery life (around 11 hours).'

15. P6, last line – suggest '... had been REMOVED by prisoners ...'.

Amended as suggested, now reads:

'as air quality monitors had been removed by prisoners during earlier sampling.'

16. P7 – this info re repeat data collection probably belongs earlier.

Have now moved to earlier in this section as suggested.

17. P9, 1st para – is Trackpro software something specific to SidePak? And re '... monitors that appeared to have been moved or tampered with ...' – I'm wondering if this needs a sentence in data collection – who checked this – was it the researchers returning and finding the monitor not quite how they'd left it?

Trakpro software comes free of charge with the Sidepak monitors, and is recommended by the manufacturers for downloading data from the monitors. Amended wording to read as follows:

'Each dataset was downloaded from the Sidepak device using the monitors recommended software (Trakpro 4.6.1)'

Added information on who checked for moved or tampered monitors:

'Researchers worked as a pair, with one completing the sampling log sheet, whilst the other positioned or retrieved the monitors and checked if they had been tampered with or moved.'

18. P9, 1st para – were Wilcoxon's tests (ie non-parametric) used because the data were skewed?

Yes, this test was used as the data is skewed.

19. P9, 1st para – typo – '... alongside ITS unique code'.

Amended typo, and in the following sentence:

'imported into STATA 13, alongside its unique code. Datasets were then paired using its unique code..'

RESULTS

20. P10, 1st para – '... collected from 29 wing landings ...' - Table 2 shows total wings as 32 (with two closed on one prison, a bit unclear whether these are included on the table), so does this mean that data couldn't be collected from three wings? And it's only at this stage, when I read about 113 datasets before and 74 after that I realise that these represent several days in all/most wings.

This is an error/typo, all wings at all four prisons (32) were sampled pre-policy, however due to two wing closures at HMP4 and one wing at HMP 3 not being accessible to researchers (due to prisoner

incident) post-policy, paired data is only available from 29 wings. Now amended this section to reflect this. Now reads as follows:

'A total of 200 datasets were collected from 32 wing landings locations throughout the four prisons. One SidePak monitor was destroyed during pre-implementation data collection'

'We therefore generated 74 paired sets of data for analysis across 29 wing landings (post-policy two wings at HMP 4 and one wing at HMP 3 could not be accessed due to security concerns).'

21. P10, 1st para – which prison had the CSU?

All four prisons had a care and separation unit have added this under 'Study prisons' section

'All four prisons had a Care and Separation Unit.'

Therefore the sentence in the results should be plural. Amended as follows:

'On sampling days both pre- and post-policy implementation all wings, apart from the Care and Separation Units, were at or near full capacity, with prisoner occupancy per wing ranging from 19 to 180.'

22. P11 – Table 2 – I don't think the 51%/11% in the final row can be correct – this is adding the other percentages, but shouldn't it be their average? Otherwise, if the others were all – say – 30%, by this logic, this row would read 120%, which doesn't make sense.

The combined 'Percentage of time above 25 µg/m³' is correct at 51%/11%. However, the 'Percentage of time above 25 µg/m³' for the four individual prisons is incorrect. Apologies, this is a data error on our part, with percentages originally being calculated from the whole data set oppose to the four individual datasets. Amended figures now entered into Table 2 as below. These figures now correspond with the median scores for each prison, for example, if the median PM2.5 for a prison is over 25 then this would suggest that over 50% of the time should be over the WHO guidance of 25 µg/m³.

	Combined prison data		HMP 1		HMP 2		HMP 3		HMP 4	
Percentage of time above 25 µg/m³ ^a	51%	11%	77%	13%	7%	0%	53%	14%	56%	15%

The text relating to this data has also been amended.

'Apart from HMP 2 (where monitors were not placed directly on the wing landings), the other three prisons lowered the time spent over the WHO 24- hour mean PM_{2.5} upper limit from 53-77% to 13-15%.'

23. P12 – 'Excluding HMP 2 ..' – but should it be excluded? Isn't this a bit like saying 'excluding the one which already happened to be low, the others reduced', but HMP 2 is as much part of the sample as the others.

Now removed the word 'excluding' from this sentence to make it clear than no data was removed. This was simply to illustrate that all but HMP 2 had similar reductions in PM2.5, and that HMP 2 was

where there were issues with carrying out direct samples from the prison wing landings. This is then addressed in the discussion section.

Results

Apart from HMP 2 (where monitors were not placed directly on the wing landings), the other three prisons lowered the time spent over the WHO 24- hour mean PM_{2.5} upper limit from 53-77% to 13-15%.

Discussion

'Since security concerns and the design of the wings at HMP 2 and HMP 3 required us to place the Sidepak monitors in wing offices (not directly on the wing landing) and cupboards on the landings (which inhibited air flow), these measures are likely to have underestimated the true PM_{2.5} concentrations on these wing locations pre- and post- smoke-free policy. Nevertheless, reductions in PM_{2.5} concentrations were still observed after policy implementation in the majority of these samples.'

24. P12 ref to Fig 2 – unfortunately the reproduction in the paper is tiny, with poor resolution, so I couldn't check this.

Figures 1 and 2 are currently provided as per requested under BMJ formatting guidelines (<https://authors.bmj.com/writing-and-formatting/formatting-your-paper/>)

Currently in TIFF format with resolution of 600 dpi. Please could BMJ Open advise if this format needs amending.

DISCUSSION

25. P14, 1st para – '... half of all sampling time ...' – think this relates to the 51% as per my comment above, so amend if this is a mistake.

Please see response to comment 22, this sentence is correct to read 'half of sampling time'.

26. P15, 1st para – '... limited by regime changes or episodes of lock-down due to prisoner incidents, which resulted in fewer samples after the policy ...' – this seems to be implying that there were more incidents post implementation – is this correct? Is it an issue/worth commenting on? Re 11 hour sampling limitation – still unclear – did this mean that researchers put it somewhere then returned the next day (by which time battery flat) to replace battery and move it? And given the various limitations listed, I'm wondering whether saying the data gives a 'VERY good estimation' might be a slight over-statement?

No, we think this would be an unfair assumption, as we worked around regime changes/episodes of lock-down/prisoner incidents pre-policy (for example, if one wing was not accessible to sample on a certain day we could re-visit it the next day to sample pre-policy). Post-policy we did now have this flexibility as we wanted to repeat the wing/day/time of sampling pre-policy. Therefore we were hindered if access to a certain wing was denied due to regime changes/episodes of lock-down/prisoner incidents post-policy. This section has now been changed to reflect this:

'Pre-policy, researchers were able to work around any prison incidents (e.g regimes changes, prisoner disturbances) in an attempt to sample all wing locations throughout the four prisons. Post-policy, researchers did not have the same flexibility as the sampling schedule was predetermined (in order to pair the samples for prison, day of the week, time, wing and monitor location) therefore fewer datasets were collected.'

See comment 12 under the Methods section which addresses the 11 hours of sampling time and how monitors were used to sample throughout the day.

Given that over 15 days worth of data was comparable pre- to post policy and this data was paired for day of the week, wing, monitor location and time, authors would still argue that this does give a very good estimation of the reduction in SHS levels after a prison smoke-free policy. Especially when compared to international air quality studies looking at changes in SHS levels after a prison smoke-free policy. This sentence reads:

'However our data does give a very good estimation, in view of the large amount of paired data collected pre- and post- policy (over 15 days pre- and post- policy), of SHS pollution during times when non-smokers would be exposed during waking hours.

27. P16, 2nd para – typo – 'found that prisons WHICH implemented..'

Amended as suggested, now reads as follows:

'This study found that prisons which implemented a smoke-free policy..'

28. P16, 2nd para, final sentence – suggest '... roll-out of A COMPREHENSIVE smoke-free POLICY across and the 32,000 staff ...'

Agreed, changed to read as suggested:

'With these findings in mind, the future roll-out of a comprehensive smoke-free policy across all 121 prisons in England and Wales set out by HMPPS (15) has the potential to have the same positive health impact on the nearly 83,000 prisoners (29) currently held, and the 32,000 staff members employed (30), in the prison estate.'

29. P16, final para – 0-3073 – is this high figure correct? The box plots suggest this would be a massive outlier.

Yes this is the correct figure and it was a outlier. For the purposes of the boxplot (Figure 1) and ease of representation, it does not show the very extreme outliers (PM_{2.5} concentrations over 500 µg/m³ which were only recorded in two HMPs). This information is outlined under the boxplot figure.

30. P17, final sentence – 'evidence in support of extending smoke-free policy throughout the E & W ...' – same comment as on this point in the strengths/limitations bullets.

Have now amended this sentence to reflect this comment, now reads as suggested, that the study supports the continued implementation of the policy.

'However our data provide strong evidence in support of the continued implementation of the smoke-free policy throughout the English and Welsh prison estate and in other penal systems internationally.'

VERSION 2 – REVIEW

REVIEWER	Sam McCrabb University of Newcastle, Australia
REVIEW RETURNED	22-Jan-2019

GENERAL COMMENTS	 1. Add 'prisoner' to this sentence: "The exemption allowed prisoners aged over 18 years to smoke in a single cell or in a cell shared with other smokers [5], staff smoking was prohibited within prison perimeter walls" 2. Update "of 60 one seconds sample measurements" to "of 60 one second sample measurements"
--

	3. What was the prisoner roll count post policy implementation? (table 1) Changes in air particulate may also be due to changes in prisoner count so would be good to add this information if possible to table 1.
--	--

REVIEWER	Helen Sweeting University of Glasgow
REVIEW RETURNED	25-Jan-2019

GENERAL COMMENTS	The authors have done a great job of responding to the reviewer suggestions. This is now SO MUCH clearer, and so also demonstrates the rigour of the study much better than the original version (eg p7, new section on 'Two researchers...'). My comments are now all very small, mainly just picking up typos. In addition, the reviewer checklist has reminded me I can't spot a statement re ethics – should this be included? Introduction P4 (first page of intro on my pdf), line 5 – typo – 'the exemption allowed PRISONERS to ...' P4, line 41 – should this be really clear that it prohibited possession AND SMOKING of tobacco – or is it completely obvious, since you can't smoke what you don't possess. P4, final para – should this include some reference to the policy on e-cigarettes or RVDs during the time-period under consideration? Methods P6, line 37 – think 'Where possible ...' should be the start of a new sentence. P7, line 48 – typo, dataset shouldn't be plural. P7, line 50 – typo, apostrophe needed in machine's. P7, line 57 – suggest 'placed' rather than 'sat'. P8, lines 10-12 – suggest '... monitors IN THIS PRISON were placed at one end of the unit ... directly taken from the wing landings IN HMP 2'. P8, line 21, typo – no S' in patient. P10, line 5 – typo – monitors needs apostrophe. P10, line 10 – typo – '... using THEIR unique CODE'. Results P11, line 33 – typo – needs a coma after 'implementation'. P11, line 55 – does 'more than half' refer to the 51% of the time? If so, would it be better to describe this as 'half' or 'just over half'?
--

	P14, line 20 – I know I suggested not using ‘excluding’ – but I’m also wondering about ‘apart’. Could you just say something like ‘In HMP 2 (where monitors were not placed directly on the wing landings) the time spent over the WHO 24- hour mean PM_{2.5} upper limit reduced from 7% to 0%; in the other three prisons, the reduction was from 53-77% pre- to 13-15% post-policy implementation’. Discussion P18, line 51 – suggest ‘... comprehensive PRISON smoke-free policies’ for clarity. P19, line 21 – ‘... the future roll-out ...’ – needs amending because all E&W prisons were smoke-free by the end of 2018. Fig 2 is still too small in the pdf to allow me to check it properly. STROBE checklist – presumably ‘ENTER TITLE ONCE DECIDED’ needs amending.
--	---

VERSION 2 – AUTHOR RESPONSE

Reviewer: 2

Reviewer Name: Sam McCrabb

Institution and Country: University of Newcastle, Australia. Please state any competing interests or state ‘None declared’: None declared

1. Add ‘prisoner’ to this sentence: “The exemption allowed prisoners aged over 18 years to smoke in a single cell or in a cell shared with other smokers [5], staff smoking was prohibited within prison perimeter walls”

Thank you for spotting this, typo, now amended to read as follows:

‘The exemption allowed prisoners aged over 18 years to smoke in a single cell or in a cell shared with other smokers’

2. Update “of 60 one seconds sample measurements” to “of 60 one second sample measurements”

Thank you for spotting this. Amended as follows:

‘60 one second sample measurements’

3. What was the prisoner roll count post policy implementation? (table 1) Changes in air particulate may also be due to changes in prisoner count so would be good to add this information if possible to table 1.

Now added in the pre-policy prisoner roll count to Table 1.

Prisoner roll count post-policy	477	628	691	378~
-----	-----	-----	------

Reviewer: 3

Reviewer Name: Helen Sweeting

Institution and Country: University of Glasgow, UK Please state any competing interests or state 'None declared': None declared

The authors have done a great job of responding to the reviewer suggestions. This is now SO MUCH clearer, and so also demonstrates the rigour of the study much better than the original version (eg p7, new section on 'Two researchers...'). My comments are now all very small, mainly just picking up typos. In addition, the reviewer checklist has reminded me I can't spot a statement re ethics – should this be included?

Ethical statement has been included, it is at the end of the Methods section and has also been completed as part of the online submission questions.

In the body of the paper, as follows:

Ethics approval

The University of Nottingham, Medical School Ethics Committee (G06062013 CHS EPH) approved this study, it was then subsequently approved by NOMS, National Research Committee (NRC) (Ref: 2013-202) in July 2014. Permission to enter all four prisons for data collection was sought from the Deputy Director of Public Sector Prisons and the Deputy Director of Custody for the South-West area. The Governors at each prison also agreed to the research being undertaken at their establishments.

Introduction

P4 (first page of intro on my pdf), line 5 – typo – 'the exemption allowed PRISONERS to ...'

As above in Reviewer 2 comment. Thank you for spotting this, typo, now amended to read as follows:

'The exemption allowed prisoners aged over 18 years to smoke in a single cell or in a cell shared with other smokers'

P4, line 41 – should this be really clear that it prohibited possession AND SMOKING of tobacco – or is it completely obvious, since you can't smoke what you don't possess.

Very true, however maybe it is not completely obvious. I've amended the wording to add 'smoking' to make this clearer. Amended as follows:

'This policy prohibited all staff members and prisoners from smoking tobacco and possessing tobacco or smoking paraphernalia...'

P4, final para – should this include some reference to the policy on e-cigarettes or RVDs during the time-period under consideration?

At this time the only policy regarding e-cigarettes was that they were purchasable through the prison canteen. As this paragraph states:

'...and disposable electronic cigarettes were made available to purchase through the prison canteen'.

Now added a sentence to state where the disposable electronic cigarettes could be used in the prison, as follows:

'Prisoners were only permitted to vape whilst in their cell.'

Rechargeable vaping devices (RVDs) were introduced after the four pilot sites went smoke-free (and after the air quality sampling taken 3 months post-policy). The introduction of RVDs was a recommendation taken forward from the four pilot sites.

Methods

P6, line 37 – think 'Where possible ...' should be the start of a new sentence.

Amended as suggested:

'Data were collected over three or four consecutive weekdays before the smoke-free policy was introduced. Where possible, data collection was then repeated after the policy was introduced...'

P7, line 48 – typo, dataset shouldn't be plural.

Amended as suggested:

'The sampling duration of each dataset was determined by access to wings locations via the prison escort and the machines battery life..'

P7, line 50 – typo, apostrophe needed in machine's.

Amended as suggested:

'...locations via the prison escort and the machine's battery life..'

P7, line 57 – suggest 'placed' rather than 'sat'.

Amended as suggested:

'Where possible, monitors were placed in discreet static locations..'

P8, lines 10-12 – suggest '... monitors IN THIS PRISON were placed at one end of the unit ... directly taken from the wing landings IN HMP 2'.

Amended as suggested:

'..all monitors in this prison were placed at one end of the unit next to or inside..'

P8, line 21, typo – no S' in patient.

Amended as suggested:

'There was no patient or public involvement in this study.'

P10, line 5 – typo – monitors needs apostrophe.

Amended as suggested:

'Each dataset was downloaded from the SidePak device using the monitor's recommended software.'

P10, line 10 – typo – '... using THEIR unique CODE'.

Amended as suggested:

'Datasets were then paired using their unique code'

Results

P11, line 33 – typo – needs a comma after ‘implementation’.

Amended as suggested:

‘On sampling days both pre- and post-policy implementation, all wings’

P11, line 55 – does ‘more than half’ refer to the 51% of the time? If so, would it be better to describe this as ‘half’ or ‘just over half’?

Amended as suggested:

‘...and continuously monitored levels were above this limit for half of all sampling time’.

P14, line 20 – I know I suggested not using ‘excluding’ – but I’m also wondering about ‘apart’. Could you just say something like ‘In HMP 2 (where monitors were not placed directly on the wing landings) the time spent over the WHO 24- hour mean PM_{2.5} upper limit reduced from 7% to 0%; in the other three prisons, the reduction was from 53-77% pre- to 13-15% post-policy implementation’.

Amended as suggested:

‘In HMP 2 (where monitors were not placed directly on the wing landings) the time spent over the WHO 24- hour mean PM_{2.5} upper limit reduced from 7% to 0%; in the other three prisons, the reduction was from 53-77% pre- to 13-15% post- policy implementation.’

Discussion

P18, line 51 – suggest ‘... comprehensive PRISON smoke-free policies’ for clarity.

Amended as suggested:

‘shown that comprehensive prison smoke-free policies are effective in substantially reducing SHS’

P19, line 21 – ‘... the future roll-out ...’ – needs amending because all E&W prisons were smoke-free by the end of 2018.

Amended as suggested and removed the word ‘future’:

‘With these findings in mind, the roll-out of a comprehensive smoke-free policy across all 121 prisons in England and Wales set..’

Fig 2 is still too small in the pdf to allow me to check it properly.

I will contact the journal about this, as the pdfs have been provided in the format they recommend.

STROBE checklist – presumably ‘ENTER TITLE ONCE DECIDED’ needs amending.

Amended as suggested, now added name of paper in.